# Feasibility Synthesis and Characterization of Gadolinia Doped Ceria Coatings Obtained by Cathodic Arc Evaporation

**DOI:** 10.3390/nano11051211

**Published:** 2021-05-03

**Authors:** Pascal Briois, Eric Aubry, Armelle Ringuedé, Michel Cassir, Alain Billard

**Affiliations:** 1FEMTO-ST, UMR 6174, CNRS, Universite Bourgogne Franche-Comté, UTBM, Site de Montbéliard, F-90010 Belfort, France; eric.aubry-01@utbm.fr (E.A.); alain.billard@utbm.fr (A.B.); 2FC Lab Research (FR CNRS 3539), Rue Thierry Mieg, F-90010 Belfort, France; 3Chimie ParisTech, PSL University, CNRS, Institut de Recherche de Chimie Paris, 11 rue Pierre et Marie Curie, F-75005 Paris, France; Armelle.ringuede@chimieparistech.psl.eu (A.R.); michel.cassir@chimieparistech.psl.eu (M.C.)

**Keywords:** gadolinia doped ceria, thin film, arc cathodic

## Abstract

Gadolinia doped ceria coatings were elaborated by cathodic arc evaporation from a metallic Ce–Gd (90–10 at.%) target inserted into a conventional multiarc Ti evaporation target in the presence of a reactive argon–oxygen gas mixture. The structural and chemical features of these films were determined by x-ray diffraction and scanning electron microscopy. Their electrical properties were characterized using impedance spectroscopy measurements. It was shown that the as-deposited coatings crystallize in the fluorite type fcc structure of ceria and that their composition is the same as that of the target. The morphology of the coatings is influenced by the evaporation parameter (stress and droplet). The electrical measurements showed two contributions in Nyquist representation and the activation energy was slightly higher than that given in the literature data for the bulk material.

## 1. Introduction

According to the Arrhenius law, the ionic conductivity of solid electrolytes is thermally activated and the conventional material used for oxygen transport in numerous applications such as solid oxide fuel cells (SOFC) is yttria stabilized zirconia (YSZ) [1,2]. The ionic conductivity increases with the operating temperature, but high temperatures are the origin of several problems: high reactivity between each component of the cell core yielding to the formation of insulating phases, high cost of the ceramic components, and brittleness of the stack due to both the high temperature and the discrepancy of the thermal expansion coefficients (TEC) of those constituting parts [1,2,3,4,5]. It is thus necessary to decrease the operating temperature of SOFC without a deterioration in their performances, which requires several adaptations to limit the loss of ionic transport properties of the electrolyte: the first is the use of new ionic conductors more efficient than YSZ, and the second is the elaboration of the stack in thin film to decrease the electrical resistance of each element. Many oxide ion conductors have been found to be potential candidates to replace YSZ: ceria-based oxide with fluorite structure [5,6,7,8], perovskite-related phases based on LaGaO_3_ [8,9], Ba_2_In_2_O_5_ [10], Bi_4_V_2_O_11_ (BIMEVOX) [8,11], La_2_MoO_9_ (LAMOX) [12] derivatives, or apatite structure lanthanum silicate [13,14]. In these materials, oxygen conduction takes place via a vacancy mechanism and, as a consequence, the conductivity strongly depends on vacancy concentration and oxygen mobility, which is then promoted by high temperatures. Ceria-based materials are serious candidates with samarium [5,6,7,15,16,17] or gadolinium [5,6,17,18,19] substitution to increase the vacancy concentration. Several articles have underlined the available techniques leading to the synthesis of gadolinium doped ceria (GDC) coatings by means of conventional chemical routes (solid state reaction [20], sol-gel [21], sintering [22], pechini [23]), and physical methods (atomic layer deposition [24], electron beam [25], reactive DC magnetron sputtering [26,27] or RF magnetron sputtering [25,28]). As far as we know, no study investigating the cathodic arc evaporation method has been reported in the literature. Indeed, cathodic arc evaporation is a powerful technique for high-rate deposition of numerous metallic alloys and more or less complex ceramic coatings, mainly carbides and nitrides [29], but very few works have dealt with the deposition of oxides [30,31].

In this paper, we present some recent results about GDC coatings synthesized by cathodic arc evaporation from a composite metallic target constituted by a cerium–gadolinium (90–10 at.%) disc inserted in a conventional multiarc titanium target under Ar–O_2_ reactive gas mixture. After a brief description of the experimental device, the structural and chemical characterizations are driven as a function of the evaporation parameters. Finally, the electrical properties of thin films are investigated by impedance spectroscopy as a function of the temperature.

## 2. Experimental Details

Ce–Gd–O films were deposited on stainless steel (AISI430) substrates by cathodic arc evaporation under reactive argon–oxygen gas mixtures of a composite target constituted of a 50 mm diameter Ce–Gd (90–10 at.%) metallic disc inserted into a conventional multiarc Ti evaporation target (Figure 1). The reactor is a 100 L stainless steel cylinder pumped down via a secondary oil diffusion vacuum system, allowing a base vacuum of about 10^−4^ Pa before refilling with argon (20 sccm) and oxygen (30 to 50 sccm). The arc discharge characteristics were 60 A and about −30 V, yielding a total pressure during the run in the range of 0.4–0.6 Pa, depending on the inlet oxygen flow rate. The AISI 430 stainless steel substrates (20 × 20 mm^2^) were placed next to the target at a draw distance of 180 mm. Prior to the deposition phase, they were polished with diamond paste (3 µm), degreased under ultra-sound in a hot dichloromethane bath, rinsed with alcohol, and dried in hot air. An in situ ion cleaning of their surface was realized in pure argon for 1 min by biasing them (V_S_ = −1000 V) in the presence of the arc discharge and then, by biasing them at −400 V during the first five minutes of the deposition stage. The following of the run was realized with a substrate bias of −50 V after oxygen introduction at the desired flow rate.

The structural features of the coatings were recorded by glancing angle x-ray diffraction using a BRUKER ((Karlsruhe, Germany) D8 focus diffractometer (Co K_α1+α2_ radiations) equipped with a LynxEye linear detector with a fix incidence of 4°. Diffractograms were collected under air flow for 15 min in the 20–80° scattering angle range by steps of 0.0019°. The chemical compositions were estimated by using energy dispersive spectroscopy (Quantax Bruker with XFLASH 6|30 detector, Bruker nano, Berlin, Germany)) coupled with a field emission scanning electron microscope (JEOL JSM-7800 F, Croissy sur Seine, France) using an accelerating voltage of 15 kV and a probe current of 2 nA. The morphology of the coatings was observed on the surface via the same SEM.

Electrical measurements were carried out using a frequency-response analyzer, PGStat30 Autolab Ecochemie BV (Metrohm France, Villebon-sur-Yvette, France). A two-electrode configuration was used to realize the impedance spectroscopy measurements. The metallic substrate (stainless steel) constituted the first electrode and a spiral of platinum wire the second. The platinum electrode geometric surface was 0.26 cm^2^. In order to separate the electrical contribution of the thin layer from the electrode response, the a.c. signal amplitude was varied from 100 to 300 mV. The measurements were realized from 1 MHz to 1 Hz, using 11 points per frequency decade. All measurements were carried out under atmospheric pressure as a function of temperature from 200 to 560 °C. The impedance diagrams were deconvoluted using the fitting software EQUIVCRT, commercialized by Boukamp [32] and more details on the calculation of electrical parameters (equivalent capacitance or relaxation frequency) have been given in a previous paper [33]. Due to the non-symmetrical two-electrode configuration, the analytical calculation of the conductivity from the measured resistance was not possible. Thus, only resistances are discussed in the paper and not the conductivity. In terms of geometrical parameter, the thickness of coating and the surface of the electrodes were kept constant.

## 3. Results and Discussion

The main deposition conditions used in this study are summarized in Table 1. Although the EDS technique does not allow for good precision of the oxygen content in the coatings, it gives a convenient estimation of the metal composition and then of the Ce/Gd ratio. Figure 2 represents the interesting range of EDS measurements for the Ti, Ce, and Gd emissions lines. With the precision of the EDS technique on oxides, this ratio systematically presented a quite constant value of around 8.2 to 8.9 (see the table insert in Figure 2), corresponding to that of the target (value 9). The Ti content was lower than 0.7 at.% in the interaction volume, meaning that the Ti pollution in the coating was insignificant, showing the confinement of the arc spot over the Ce/Gd surface (even no magnetic field). The thickness of the coatings were estimated by the SEM observations from the rupture facies. Unfortunately, the irregular coating morphology intrinsic to the cathodic arc processes and the crystallographic nature (fcc) of the substrate with slip planes led to a ductile rupture rather than an expected fragile rupture. Then, the average thickness of the coating was estimated at about 650 ± 150 nm. XRD performed on the as-deposited coating showed that all coatings crystallized under the expected fluorite fcc structure of ceria (Figure 3). However, a slight displacement of the diffraction peaks toward lower Bragg angles was observed for the thinner coatings (GDC09 and GDC11: about 500 nm-thick, see Table 2), which was also less crystallized than the thicker ones (GDC08 and GDC10: about 700 nm-thick see Table 2). Knowing that the oxygen flow is sufficient to synthesize a stoichiometric compound (reactive sputtering mode), the observed shift of the diffraction line would then be rather induced by the formation of growth stress and/or an increase of the cell parameter.

The lattice parameter for each layer (Table 2) was determined from the diffraction plane (111), which changed from 0.542 to 0.558 nm. These values were close to the theoretical parameter (0.5418 nm from the JCPDS card 75–161). These results agree with the results of J. Jiang et al. [28], who measured a lattice parameter varying from 0.543 to 0.545 nm for a coating synthesized by RF magnetron sputtering. According to our results, the lattice parameter increases with stresses as well as the reduction in thickness. The crystallite size was estimated with the Scherrer method by considering the spherical crystallites independent of the crystallographic directions and by assuming a negligible instrumental contribution to the line broadening compared to that induced by the size of the crystallites. The grain size of the coatings varied from 6 to 13 nm, in agreement with the study of Y.S. Hong [25], who determined a crystallite size of 7 nm for GDC films produced by electron beam-PVD, but was lower than that measured for sputter-deposited coatings (38 nm) in our previous study [33]. It is noteworthy that the GDC08 coating exhibited a relatively pronounced preferential orientation following (111) directions. This phenomenon is not well understood, however, it can be clearly attributed to the arc spot mobility favored by the placement of two magnets behind the target (see Table 1).

Figure 4 shows the top surface micrographs observed by SEM. All films presented a dense morphology with the presence of more or less droplets, which is typical of a coating obtained by cathodic arc evaporation. The droplet concentration is driven by the presence (or not) of magnets behind the cathode; this concentration decreases when the magnetic field increases. Indeed, macroparticle emission is known as a function of the spot mobility [34], which increases with the intensity of the magnetic field. Without the magnetic field, the spot path is random, and its mobility is low. Hence, the volume of molten metal is high as well as the ability of macroparticle ejection [35]. When applying a magnetic field (B) parallel to the surface of the target and then perpendicular to the electric field (E), the spot is submitted to a rotating movement following a direction opposite to that predicted by the Lorentz force ( F→=q∗J→^B→ ) and called a “retrograde movement” [36]. Hence, coatings deposited with magnets behind the target (steered arc) usually present less macroparticles than those deposited without magnets (random arc). Figure 4 also shows the importance of the oxygen partial pressure on the film quality. Indeed, the rate of macroparticle emission is also ascribed to the melting temperature of the target surface: as with magnetron sputtering, a low oxygen flow rate induces an evaporation in the so-called elemental or transition modes (i.e., with a mainly metallic target surface), whereas high oxygen flow rates yield a whole poisoning of the target, which operates in the so-called reactive evaporation mode. If elemental or transition modes are preferred in magnetron sputtering due to the deposition rate [37], the poisoning resulting from evaporation in the reactive mode increases the melting temperature of the target surface and hence decreases its ability to produce macroparticles [35]. The favorable effect of the surface poisoning of the target is clearly preponderant by considering the surface aspects of GDC10 and GDC11 coatings, elaborated with the same magnetic field but with different oxygen flow rates. The observation of the top-surface of all coatings also showed a flake aspect clearly induced by the presence of strong compressive stress in the thin films due to the rather high energy of impinging evaporated species in arc evaporation. As Pierson et al. [38] state, this particular phenomenon corresponds to successive accumulations and relaxations of stress during the deposition stage due to the high energy of impinging species and flaking, respectively.

Electrical properties of the coatings were determined by impedance spectroscopy. On the Nyquist diagram presented in Figure 5, three domains can be distinguished, which correspond to the tendency for the samples GDC09, GDC10, and GDC11, whereas only two domains were detected for GDC08. In all cases, the signal for the high and middle frequency domains did not depend on the a.c. signal, in contrast to the low frequency range. This allowed for the discrimination of the layer response from that of the electrode as it is well known that the electrode response depends on the amplitude of the a.c. signal [33]. According to Suzuki et al. [39], the behavior of GDC08 is in accordance with that of thin layers or nanostructured materials. The other coatings presented a similar behavior as that of the bulk solid electrolytes [40], with two contributions for the electrolyte response. The relaxation frequency is a parameter that does not depend on the geometric factor [41]. The electrolyte in GDC was modeled with two R//CPE circuits in series (Bulk and grain boundaries contribution); the values of the fit results are reported in Table 3 for each sample as a function of the temperature.

In Figure 6, we compare the relaxation frequencies of the arc evaporated GDC coatings to those measured on a bulk 3.6 mm thick GDC whose signature for bulk and grain boundary responses was more pronounced [33]. For arc evaporated coatings, the first contribution was slightly higher than the second one, but all values remained lower than those obtained with the pellet, which could be due to the smaller grain size of the coating. Another electrical parameter can be used to estimate if both contributions correspond to the bulk and the grain boundary responses: the equivalent capacitance. The equivalent capacitances as a function of the temperature for the different coatings of this study were quite constant (Figure 7). The first and second contributions were around 10^−11^ and 10^−9^ F, respectively. According to Bauerle [40] and Tschöpe et al. [41], impedance spectroscopy measurements performed on cathodic arc evaporated coatings are quite similar to those performed on bulk ceramic sintered pellets, with two contributions for the electrolyte (i.e., 10^−12^ and 10^−9^ F for the bulk and the grain boundary contributions 1 and 2, respectively). 

The coating resistance as a function of the temperature is presented in Arrhenius representation in Figure 8. From its linear evolution, it can be concluded than the oxide ion conduction mechanism is thermally activated, as expected. All activation energies measured from Figure 8 are summarized in Table 4. For a bulk GDC pellet, the activation energy of the bulk response was slightly lower than that of the grain boundaries [42]. Moreover, the resistance of the bulk contribution was also lower than that of the grain boundaries at a given temperature; consequently, the ionic conductivity is higher for a bulk controlled mechanism. The apparent activation energy of all the arc evaporated coatings was in the range 0.62 to 0.92 eV. Such values are in the same order of magnitude as the literature data for bulk GDC (0.89 eV [43]). Finally, except for GDC8, which exhibited only one contribution, all the coatings presented an evolution of the LnR vs. 1/T slope around 400 °C. As the conduction mechanism is expected to be less and less controlled by the grain boundaries with increasing temperature [42], it was assumed here that the transition between bulk and grain boundary conduction occurs at around 400 °C. This phenomenon supports previous studies where sputter deposited coatings exhibited a single contribution mainly driven by grain boundary conduction phenomena [33].

## 4. Conclusions

Ce–Gd–O coatings were synthesized by arc cathodic evaporation on rotating substrates from a composite target consisting of a Ce–Gd (90–10 at.%) disc inserted into a conventional multiarc Ti evaporation target in the presence of reactive argon–oxygen gas mixtures. The as-deposited coatings were crystallized under the fcc structure of ceria and presented a constant composition. The morphology of the coatings was dense with the presence of droplets. The flake morphology of the thin films was induced by the high stress level proceeding from the deposition stage. Electrochemical impedance measurements performed on the coatings presented the same behavior as that of the bulk samples; indeed, two contributions were observed for the electrolyte response (one for the bulk and the second one for the grains boundaries). The activation energy of the film was in the range 0.62 to 0.92 eV, which is in accordance with that of the literature data. Further investigations are required such as a study of the behavior of the coatings as a function of the oxygen partial pressure and synthesis on porous substrates in order to allow for their use as an electrolyte material for intermediate temperature solid oxide fuel cells.

## Figures and Tables

**Figure 1 nanomaterials-11-01211-f001:**
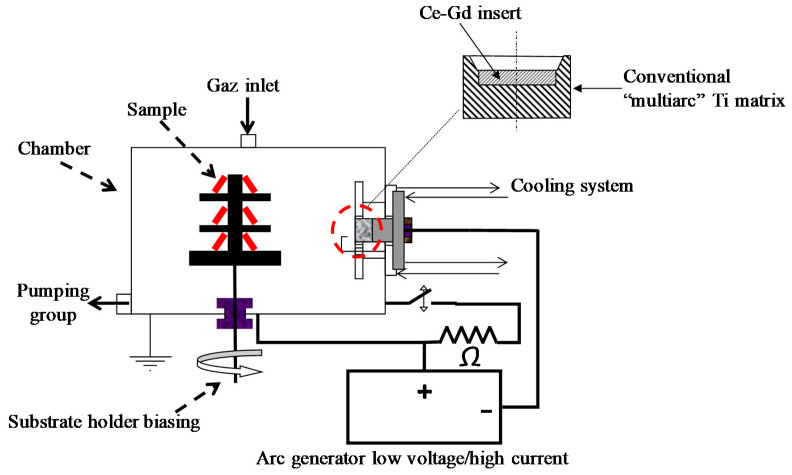
Scheme of the experimental device.

**Figure 2 nanomaterials-11-01211-f002:**
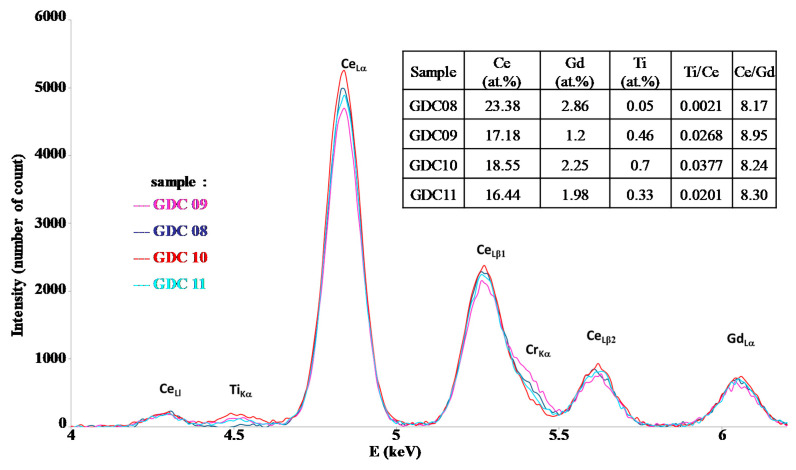
EDS spectra of GDC films deposited by the arc cathodic technique.

**Figure 3 nanomaterials-11-01211-f003:**
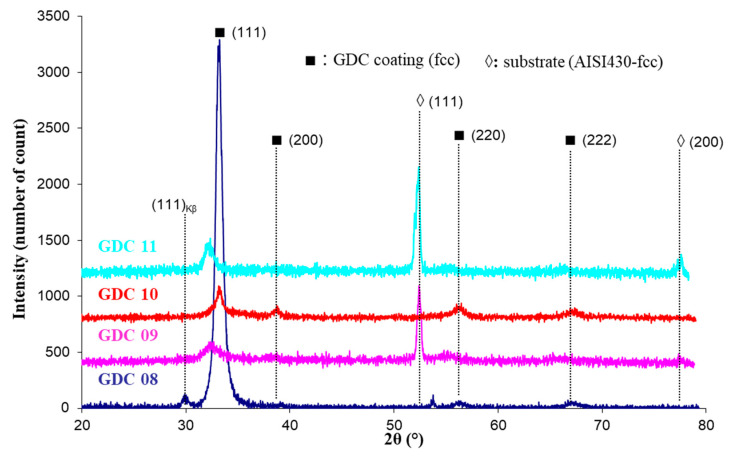
Grazing incidence X-ray diffractogram of GDC coatings deposited on AISI 430 stainless steel substrates.

**Figure 4 nanomaterials-11-01211-f004:**
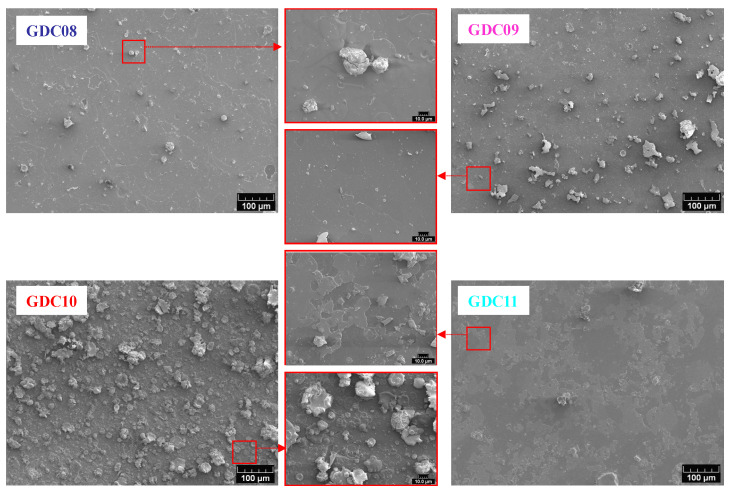
Top surface SEM micrographs of GDC coatings deposited on AISI 430 stainless steel substrates.

**Figure 5 nanomaterials-11-01211-f005:**
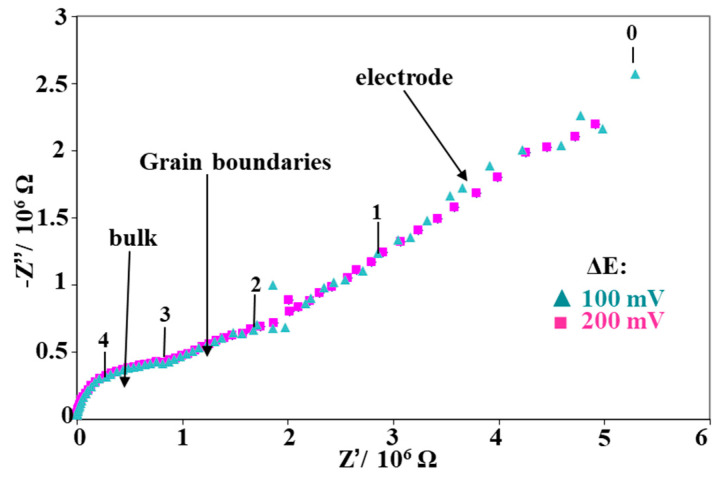
Nyquist impedance plots of the GDC09 coating, registered at 400 °C, for two a.c. signal amplitudes. The numbers 0, 1, 2, 3, and 4 correspond to the logarithm of the frequency value imposed during the impedance test.

**Figure 6 nanomaterials-11-01211-f006:**
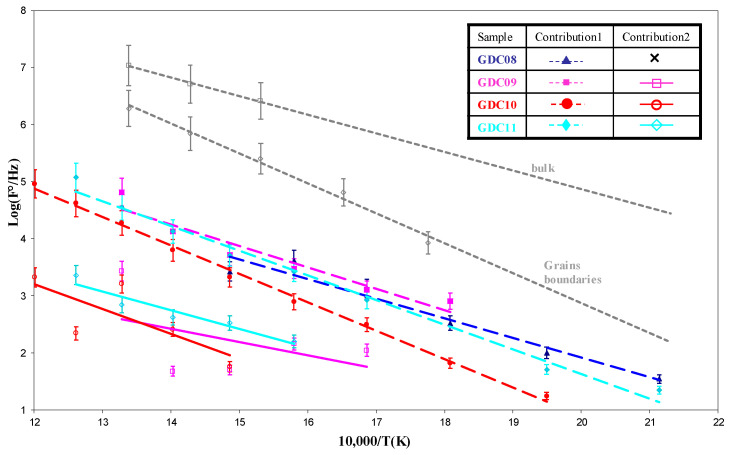
Arrhenius plots of the relaxation frequencies as a function of the temperature compared to those of a bulk sintered GDC sample.

**Figure 7 nanomaterials-11-01211-f007:**
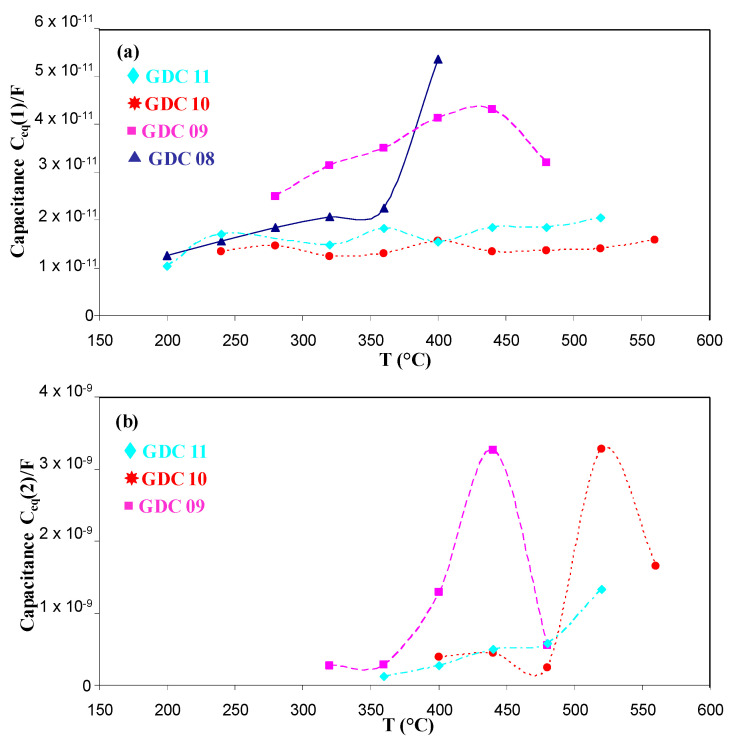
Equivalent capacitance as a function of the temperature for the first (**a**) and the second contribution (**b**).

**Figure 8 nanomaterials-11-01211-f008:**
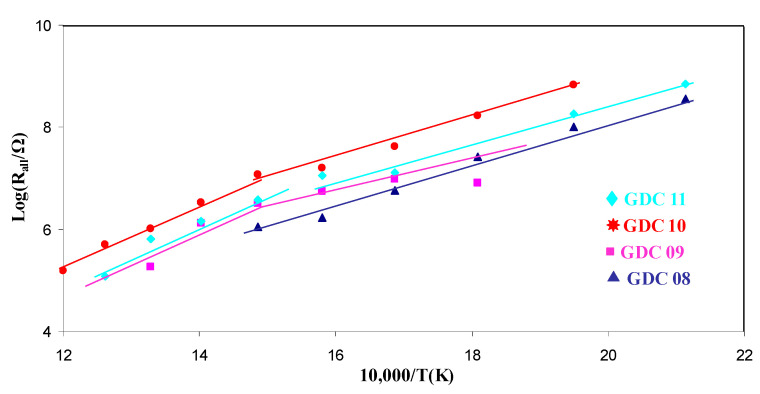
Arrhenius plots of the total resistance as a function of temperature for all GDC films.

**Table 1 nanomaterials-11-01211-t001:** Main deposition conditions of the study.

Sample	Cleaning by Bias	Bias (V/mA)	Magnet	D_Ar_ (Sccm)	D_O2_ (Sccm)	Pressure (Pa)	Intensity (A)	Voltage (V)
GDC 08	Yes	−50/500	2	20	40	0.6	60	38.5
GDC 09		−50/400	0		40	0.6		40
GDC 10		−30/700	1		30	0.4		18
GDC 11		−50/600	1		40	0.6		40

**Table 2 nanomaterials-11-01211-t002:** Lattice parameter, grain size, and thickness for each sample.

Sample	Bragg Angle (°)	Lattice Parameter (nm)	Grain Size (nm)	Thickness (nm)
GDC08	16.62	0.542	13.1	700
GDC09	16.10	0.559	5.7	500
GDC10	16.57	0.543	13.7	740
GDC11	16.17	0.556	10.4	550
Previous work [33]	16.65	0.541	38.1	1000

**Table 3 nanomaterials-11-01211-t003:** Value of the electrolyte fit result for all samples as a function of temperature.

**GDC08**	**Contribution 1 (Bulk)**					
**T(°C)**	**R (** **Ω)**	**CPE (F)**	**n**			
200	3.64 × 10^8^	1.65 × 10^−11^	0.95			
240	1.03 × 10^8^	2.89 × 10^−11^	0.9041			
280	2.63 × 10^7^	5.13 × 10^−11^	0.8657			
320	5.72 × 10^6^	8.03 × 10^−11^	0.85			
360	1.73 × 10^6^	1.03 × 10^−10^	0.85			
400	1.12 × 10^6^	3.89 × 10^−10^	0.796			
**GDC09**	**Contribution 1 (Bulk)**			**Contribution 2 (Grain Boundaries)**		
**T(°C)**	**R (** **Ω)**	**CPE (F)**	**n**	**R (** **Ω)**	**CPE (F)**	**n**
280	8.06 × 10^6^	5.33 × 10^−11^	0.91	Not detectable		
320	3.99 × 10^6^	7.05 × 10^−11^	0.91	5.40 × 10^6^	8.66 × 10^−10^	0.82
360	1.55 × 10^6^	8.45 × 10^−11^	0.91	4.00 × 10^6^	1.43 × 10^−9^	0.76
400	7.38 × 10^5^	1.59 × 10^−10^	0.87	2.48 × 10^6^	1.08 × 10^−8^	0.63
440	2.78 × 10^5^	2.96 × 10^−10^	0.83	1.03 × 10^6^	4.48 × 10^−8^	0.54
480	7.59 × 10^4^	3.20 × 10^−11^	1	1.07 × 10^5^	3.49 × 10^−9^	0.81
**GDC10**	**Contribution 1 (Bulk)**			**Contribution 2 (Grain Boundaries)**		
**T(°C)**	**R (** **Ω)**	**CPE (F)**	**n**	**R (** **Ω)**	**CPE (F)**	**n**
240	6.80 × 10^8^	1.86 × 10^−11^	0.93	Not detectable		
280	1.68 × 10^8^	2.65 × 10^−11^	0.9			
320	4.17 × 10^7^	1.79 × 10^−11^	0.95			
360	1.57 × 10^7^	2.55 × 10^−11^	0.92			
400	4.84 × 10^6^	3.34 × 10^−11^	0.92	7.05 × 10^6^	1.62 × 10^−9^	0.76
440	1.90 × 10^6^	1.65 × 10^−11^	0.98	1.40 × 10^6^	1.35 × 10^−9^	0.85
480	6.21 × 10^5^	1.36 × 10^−11^	1	4.19 × 10^5^	3.12 × 10^−10^	0.97
520	2.74 × 10^5^	1.40 × 10^−11^	1	2.20 × 10^5^	1.86 × 10^−8^	0.76
560	1.10 × 10^5^	1.58 × 10^−11^	1	4.60 × 10^4^	5.16 × 10^−9^	0.88
**GDC11**	**Contribution 1 (Bulk)**			**Contribution 2 (Grain Boundaries)**		
**T(°C)**	**R (** **Ω)**	**CPE (F)**	**n**	**R (** **Ω)**	**CPE (F)**	**n**
200	6.99 × 10^8^	1.13 × 10^−11^	0.98	Not detectable		
240	1.86 × 10^8^	2.68 × 10^−11^	0.92			
320	1.30 × 10^7^	2.63 × 10^−11^	0.9321			
360	3.34 × 10^6^	3.24 × 10^−11^	0.94	8.08 × 10^6^	3.87 × 10^−10^	0.835
400	2.03 × 10^6^	2.32 × 10^−11^	0.96	1.78 × 10^6^	2.68 × 10^−10^	1
440	6.54 × 10^5^	3.23 × 10^−11^	0.95	7.64 × 10^5^	2.40 × 10^−9^	0.8
480	2.48 × 10^5^	1.83 × 10^−11^	1	4.00 × 10^5^	3.07 × 10^−9^	0.8
520	6.59 × 10^4^	2.04 × 10^−11^	1	5.27 × 10^4^	8.97 × 10^−9^	0.8

**Table 4 nanomaterials-11-01211-t004:** Activation energy for each contribution.

Sample	Bulk (eV)	Grain Boundaries (eV)	Apparent (eV)
GDC 08	0.84	-	0.84
GDC 09	0.81	0.84	0.62
GDC 10	1.0	1.44	0.92
GDC 11	0.91	1.24	0.81

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
