# Peer review of "Feasibility Synthesis and Characterization of Gadolinia Doped Ceria Coatings Obtained by Cathodic Arc Evaporation"

_nanomaterials, 2021, doi:10.3390/nano11051211_

Round 1

Reviewer 1 Report

Dear editor, the comments and suggestions were listed in the .pdf file.

Author Response

Dear reviewer,

please find a word file in which we have answered your questions or remarks.
We remain at your disposal

Best regards,

Pascal Briois

Reviewer 2 Report

The article of P. Brois et al regarding the deposition and viability of Ce based coatings shows interesting results that might be applicable in several areas. The research is interesting and quite well structured, however, I have some concerns regarding the characterizations and some parts of the presentation of the manuscript.

My comments are as follows:

  • Please include thicknesses and grain sizes in Table 1.
  • The Gi-XRD results presented in Figure 2 show dramatic changes in intensity. The authors attributed this to the use of double magnets, however, the reason for this inclusion is not clear, as the deposition conditions change with respect to the 0 and 1 magnets. This needs to be clearly explained. Additionally, there is a peak at 30 degrees in the thick sample, that is not indexed.
  • Perhaps it would also be interesting for the study to measure the nominal changes of the (111) peak (position and strain). And support the stress related to conclusions, which seem not supported at the time.
  • Figure 4, shows the Nyquist plot of the samples. For the reviewer, it is not clear why the authors assign the regions in such a way. Authors should use an equivalent circuit to extract some information from this plot, with a clear inset of the fitted parameters and fitted curve. It is clear that low-frequency regions can be attributed to slow processes, including grain boundaries, but typically, intrinsic effects are found at high frequencies. Despite this, if authors want to follow their assumptions, two R-C (rather an R-CPE) circuits need to be fitted in series (perhaps 3 accounting for boundaries). Or other configuration, like an R-CPE, for high frequencies, followed by a Modified Randals circuit in series, to cover both the Grain boundaries as Extrinsic (low-frequency effects). In this way, Capacitance should be extracted for each independent contribution. I believe the authors did this kind of analysis, but the fitting tables are needed and the clear extracted values are also shown.
  • Temperature changes in the EIS spectra need to also be shown, with Nyquist plots also. The changes in contributions will be very clear at high temperatures and the higher dominance of some semicircles will be very clear. At this point, it is unclear if the authors contemplated this aspect.

Author Response

Dear Reviewer 2,

please find a word file in which we have answered your questions or remarks.
We remain at your disposal

Best regards

Pascal Briois

Round 2

Reviewer 1 Report

I have evaluated the revised manuscript and want to congratulate you on excellent completion of your work. Taking into account all the corrections made I recommend this manuscript for publication.

Author Response

Dear reviewer1

we thank you again for the relevance of your suggestions and your comments on our article as well as for the speed

Have a nice evening

Pascal

Reviewer 2 Report

The authors have appropriately addressed my comments. I believe the article can be accepted in its present form.

Author Response

Dear reviewer2

we thank you again for the relevance of your suggestions and your comments on our article as well as for the speed

Have a nice evening

Pascal